# Manufacturability-Based Design Optimization for Directed Energy Deposition Processes

Harry Bikas [ID], Michail Aggelos Terzakis and Panagiotis Stavropoulos *[ID]

Laboratory for Manufacturing Systems & Automation, University of Patras, 26504 Patras, Greece;
bikas@lms.mech.upatras.gr (H.B.); terzakis@lms.mech.upatras.gr (M.A.T.)
* Correspondence: pstavr@lms.mech.upatras.gr; Tel.: +30-2610-910160

**Abstract:** Additive Manufacturing (AM) is the process of joining materials by selectively depositing them layer upon layer for the purpose of manufacturing parts or assemblies which are based on a 3D digital model. The nature of these processes results in the morphing of complex component geometries, enabling a high degree of design freedom and resulting in lightweight structures with increased performance. These processes, however, experience many limitations regarding manufacturability. The aim of this study is to develop a method and tool that optimizes the design of a component to avoid overhanging geometries and the need for supports during the Additive Manufacturing process. A workflow consisting of steps for topology optimization, orientation optimization, material addition, and machine code generation is described and implemented using Rhinoceros 3D and Grasshopper software. The proposed workflow is compared to a conventional workflow regarding manufacturing Key Performance Indicators (KPIs) such as part volume, support volume, and build time. A significant reduction is observed regarding all the KPIs by using the proposed method. Examining the results from both the conventional workflow and the proposed one, it is clear that the latter has unquestionable advantages in terms of effectiveness. In the particular case study presented, a total volume reduction of around 80% is observed. The reduction in the total volume (including the required support volume) leads to a significant reduction in the material used as well as in the build time, consequently resulting in cost reduction.

**Keywords:** additive manufacturing; DED; manufacturability; topology optimization; orientation optimization; material addition; slicing; G-code

## 1. Introduction

The process of building up a part by depositing material layer by layer on a build bed or within a build chamber is known as Additive Manufacturing (AM) [1]. Due to the freedom to create complicated geometries without the use of specialized tools, AM processes introduce an entirely novel realm within engineering design and manufacturing. These advantages have made Additive Manufacturing appealing to numerous industries, including automotive, aerospace, machine tool manufacturing, healthcare, and the food industry [2]. One of the differentiating elements between the various AM process families is the process mechanism used. This is a key aspect as the process mechanism can be linked directly with the manufacturability limitations of the process. As such, categorizing AM processes according to the process mechanism is useful [3]. From the plethora of AM processes available, this work focuses specifically on the Directed Energy Deposition (DED) process family. In DED, materials are fed through a narrow nozzle and are melted during deposition using focused thermal energy. The energy source can be a laser beam, an electron beam, or a plasma arc. The deposited materials are typically metals in the form of wire or powder [4]. Powder-based DED processes typically use a laser as the thermal energy source, while wire-based DED processes utilize an electric arc, a plasma arc, a laser beam, or an electron beam as the thermal energy source [5].

This paper presents a method and software tool that enhances the manufacturability of a component for DED processes that has been generated from a topology optimization procedure by limiting and modifying the geometry of overhanging areas and thus the required support volume to be built. The goal of the presented method and tool is to improve the manufacturability of highly optimized lightweight structural parts typically designed using topology optimization, reducing the total volume of material that needs to be deposited to manufacture both the part and the support structures, thus reducing material usage, build time, and, ultimately, costs.

The emergence of Additive Manufacturing technologies and their competitive advantage, in regard to the geometrical freedom that can be obtained, have led to the increased popularity of topology optimization, a method that makes lighter and potentially equally strong components by reducing unneeded material from the initial design [6]. Different algorithmic approaches have been proposed to implement topology optimization. Bendsoe et al. [7,8] presented a homogenization method that exploits infinitely microscale holes in a design domain rather than deleting the whole of a finite element [9–13]. Solid Isotropic Material with Penalization (SIMP) is another topology optimization method that penalizes the stiffness tensor by a penalization factor (p) that helps force a solution of either a solid or a void. A penalization factor greater than one turns the intermediate densities into solid and void zones, while a penalization value equal to one will result in a density gradient [14–18]. Besides density-based approaches, researchers inspired by the nature, specifically by the optimal, organic structures, of shells, bones, and trees found that their long evolutionary period and adaption to any environment they find themselves in are the reasons for such geometries. This led them to develop a new topology optimization method, called Evolutionary Structural Optimization (ESO). The ESO approach uses a fixed model with typical finite elements to describe the initial design domain. The ultimate optimal structure is a subset of the initial set of finite elements [19–22]. Based on the ESO technique, Yang et al. [23] developed an extension dubbed bidirectional ESO (BESO) for topology optimization sensitive to stiffness and displacement limitations.

All aforementioned topology optimization strategies produce objects with unique, organic shapes aiming to be manufactured utilizing the increasingly available AM processes. Nevertheless, even the most advanced AM processes have finite manufacturing capabilities; therefore, an evaluation of the manufacturing process limitations against the part to be additively manufactured should be conducted in order to attain a high level of manufacturability. Manufacturability in general encompasses a parts' design ease of manufacturing and its potential for cost reduction [2]. Similarly, AM manufacturability is the degree to which a part that is to be manufactured using an AM process will employ the favorable properties of this given technique [24]. In order to achieve a high degree of AM manufacturability, specific Design for AM (DfAM) rules must be taken into consideration during the early design stages of a part. Bikas et al. [25] underlined the importance of these guidelines, since following them will allow the part to fulfill the standards it is designed to and attain the optimum degree of manufacturability. Furthermore, they classified them into two groups, which are the design aspects (described as any characteristic which can be measured at the design stage, such as overhangs, bores, channels, etc., as well as a part's programming parameters) and the design considerations (the consequence on the manufactured part).

One of the most essential design aspects that significantly affects the manufacturing process are the overhangs since they are directly correlated with the need to support structures [26,27]. An overhanging feature is any shape whose orientation is not parallel to the build vector, contributing to the reduction in the efficiency of the AM process build time, material, post-processing equipment, and process cost. In their work, Ghiasian et al. [28] presented a framework to evaluate manufacturing feasibility using AM considering limits imposed from parameters such as geometry, build orientation/generated supports, required time, and cost. Thereupon, numerous methods based on computational methodologies that are focusing on tackling manufacturability challenges have been offered [29–31].

Lianos et al. [32] provide a paradigm that analyzes the manufacturability of a given design considering a Directed Energy Deposition (DED) technique. Subsequently, an AM design workflow is proposed, starting with the determination of the stress situation inside the build volume of the part and then orienting in order to minimize the overhanging features and maximize its AM manufacturability.

A small amount of works have also been proposed to manipulate existing design geometries with the aim of improving manufacturability. O'Hara et al. [33] proposed three advanced mesh manipulation techniques, and post-print surface optimization can decrease design time and improve physiological simulations, aiming to help plan surgical treatments and contribute to the success of test devices.

All these approaches that address manufacturability limitations are problems that need to optimize two or more parameters simultaneously under various constraints. These problems are called constrained multi-objective optimization problems (CMOPs), and their solution depends on evolutionary algorithms [34,35].

The last piece of this process is the development of the component that occurred from the whole optimization procedure. This includes the slicing of the fully optimized mesh and the generation of the appropriate machine code (G-code or otherwise) [36,37].

After studying the aforementioned works, the conclusion is that there is no software tool available that is able to optimize the geometry of a component aiming to eliminate the areas that reduce AM manufacturability. This work focuses on addressing this gap by utilizing the significant advantages of algorithmic-aided design to propose a design strategy based on manufacturability.

## 2. Materials and Methods

The proposed method consists of 4 steps that are linked together to generate an optimal (from an AM manufacturability perspective) design. Step 1 focuses on the topology optimization process for a given boundary domain where the structural analysis is going to be performed. Step 2 presents the orientation optimization process where the optimized mesh that occurs from Step 1 is rotated around the x- and y-axis in order to minimize the overhanging areas of the mesh and thus the required support material volume. Step 3 is the material addition process, which utilizes the mesh from the previous step to create suitable geometries with the aim of removing the remaining overhanging areas that are outside the feasible limits of the used AM process. The final step slices the fully optimized mesh from Step 3 to generate the machine code to manufacture the part. The aforementioned workflow is summarized in Figure 1.

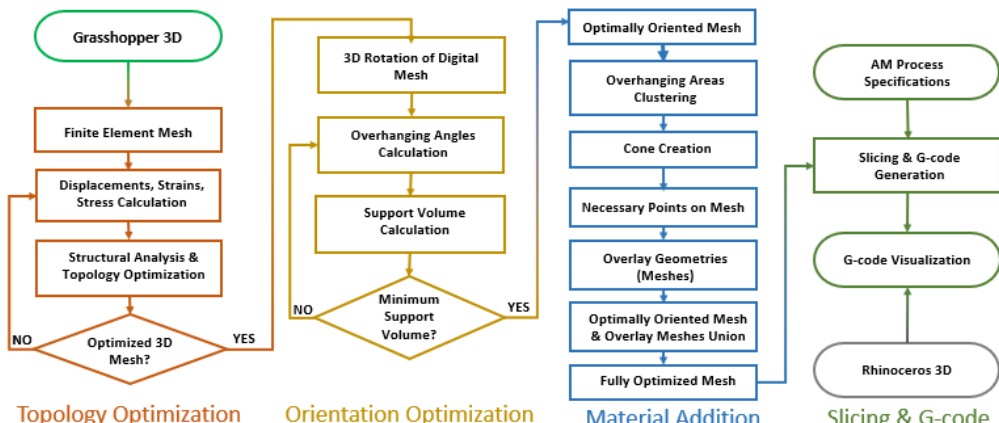

**Figure 1.** Method flow diagram.

The proposed workflow has been implemented utilizing the functionalities of the Rhinoceros 3D and Grasshopper software tools. Rhinoceros 3D is a computational-aided design-based software that uses NURBS-based tools in order to mathematically model and represent 2D or 3D geometries [38,39]. Grasshopper 3D is a visual programming language

that uses a block diagram environment so as to perform numerous functions. It is a plug-in for Rhinoceros 3D, and it has the capability to run within it and can deploy its interface in order to visualize the 3D voxelized models [40,41]. The following sections present in detail each of the aforementioned algorithm steps.

### 2.1. Topology Optimization

Step 1 of the proposed method begins by defining the domain (boundary domain) in which the 3D volumetric analysis and optimization will be executed. The domain of the problem describes the region where the topology optimization operation will occur and the final component, consisting of a 3D ISO mesh, will be produced. Following this, the areas where the forces are applied and the constraints are installed must be specified in order to perform the intended operations. Then, the maximum number of iterations that the optimization algorithm will perform (the larger the number of iterations, the better the voxelized mesh), the percentage of material that will remain inside the boundary domain at the end of the procedure (it takes values between 0 and 1), and the penalty factor that prevents the greyscale result at the voxelized component (values 2 or 3 return the best problem covering) are determined to begin the topology optimization process.

### 2.2. Orientation Optimization

Step 2 of the proposed method deals with optimizing the orientation of the designed component to minimize overhangs. The starting point of this procedure is the determination of the overhanging areas of the optimized mesh. To do that, the angle between the normal vector of each triangular mesh facet and the global horizontal plane (XY plane) is calculated [42]. If this angle surpasses the limit that the respective AM process can create without supports, then this face is characterized as overhanging geometry [43]. The required support volume for the manufacturing process is computed for every triangular facet via the projection of the area of the overhanging geometry along the XY plane multiplied by their distance from the base plane. Aiming to minimize the required support volume, the optimal combination of rotation angles around the X- and Y-axis has to be determined. Searching every possible angle combination is time-consuming (and, depending also on the step size, possibly impractical). In order to tackle this multi-parameter optimization problem, an evolution algorithm is implemented. The algorithm follows an iteration process by dividing the range of angles into smaller ranges with a higher resolution when finding a better solution each time until it reaches the optimal one. The evolutionary algorithm is implemented through a Grasshopper component called "Galapagos" [44].

### 2.3. Material Addition

The next step in the proposed method aims to cover all the remaining overhanging areas with additional material to reduce or even eliminate the need for support structures in the manufacturing process. The general idea behind the material addition procedure is based on creating a geometry that complies with the manufacturing limitations of the AM process. The basic geometrical shape that its entire surface maintains a constant angle throughout its height is a cone. Therefore, the desired overlay geometry occurs from the Boolean difference in a conical surface that has a θ-degree normal vector (where θ is the limit imposed by the AM process) with a geometry that has been created from a set of points that define the border around the problematic overhanging area. After creating the conical surface, the next step of the procedure is to find the necessary points of the part to project on the aforementioned conical surface in order to create a new geometry that will cover the overhanging areas. At first, the points of the cluster are used to find their closest points on the conical surface. Then, these points (of the conical surface) are used to find a certain number of corresponding points of the initial mesh, and these, in turn, are used to locate more points on the conical surface (Figure 2a). It has to be noted that all the overlay geometries are separate meshes and are joined into a single mesh for the next step (Figure 2b).

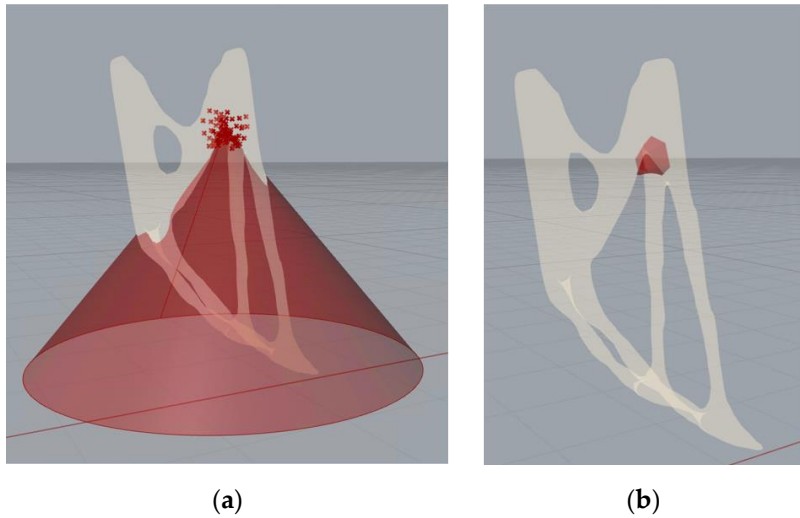

**Figure 2.** (**a**) Set of points and cone on mesh; (**b**) creation of overlay geometry on mesh.

### 2.4. Machine Code Generation

The last step of the proposed method is to slice the resulting geometry and generate the part program/machine code that is going to be used to build the part. This step is entirely dependent on the AM process that is going to be used to manufacture the optimized mesh that occurred from the previous process. In particular, parameters such as layer height, track width, deposition speed, etc., depend on the AM process and machine that will be used and determine the values that need to be entered towards generating the part program. The process parameters and variables that were used for the slicing and subsequent generation of the part program are summarized in Table 1.

**Table 1.** Process parameters.

| Parameter | Value |
|---|---|
| Layer height [mm] | 0.2 |
| Infill [%] | 30 |
| Nozzle diameter [mm] | 0.4 |
| Shell number | 2 |
| Bottom layers | 8 |
| Top layers | 0 |
| Print speed [mm/s] | 70 |
| Travel speed [mm/s] | 130 |
| Retraction speed [mm/s] | 30 |
| Retractions distance [mm] | 3 |
| Filament diameter [mm] | 1.75 |
| Flow rate [%] | 100 |

At first, the optimized mesh is sliced using an even layer height determined by the user based on the capabilities of the selected AM process. Then, the contours of the sliced mesh are utilized in order to create the infill pattern. The infill is measured as a percentage and expresses the area ratio of the layer enclosed by the contour that is going to be covered. This parameter is also determined by the user. Then, the contour and its respective infill curve are translated into points which are sorted in order to form a path that the process is going to follow to manufacture the part. The entirety of all these point coordinates, combined with the type of interpolation from point to point as well as the commands regarding the key functions of the process and machine, form the part program. This can be compiled as machine-specific code.

*2.5. Case Study*

In order to validate the proposed method, a case study of a simple cantilever beam was examined. The beam has initial dimensions of 300 × 200 × 120 (mm) for its length (*x*-axis), width (*y*-axis), and height (*z*-axis), respectively. A fixed constraint is applied in its left side, whereas a uniformly distributed linear load is applied throughout the y-axis, in an upward direction, in the lower right edge of the structure with a value of 200 N/m. (Figure 3a). The material is assumed to be 316L stainless steel with Young's Modulus E = 210 GPa and Poisson's Ratio ν = 0.3 [45]. It is worth mentioning that the properties used for the case study have been obtained from the literature and correspond to test coupons that were experimentally characterized using the same DED machine and material the authors intended to use [46]. These correspond to the properties directly obtained by the DED process without any post-processing, as this can often be the case for DED end users (contrary to the common practice of post processing in PBF processes). Since properties are dependent on the direction (and are typically lower across the deposition direction), the authors elected to use the worst scenario due to the orientation variation that characterizes the proposed method. The final material properties used are summarized in Table 2.

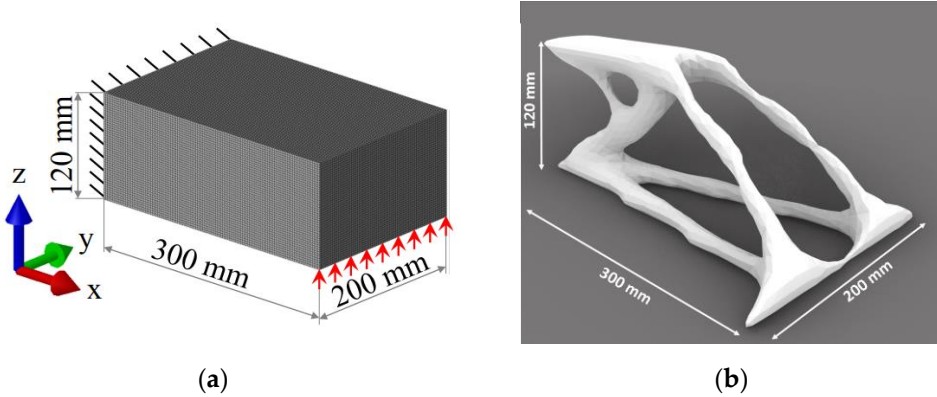

| (a) | (b) |

**Figure 3.** (**a**) Bounding box with constraints; (**b**) mesh after topology optimization.

**Table 2.** Mechanical properties of 316L stainless steel after DED process.

| Property | Meltio XY Properties | Meltio XZ Properties |
|---|---|---|
| Yield Strength [MPa] | 429 ± 6 | 347 ± 11 |
| Ultimate Tensile Strength [MPa] | 643 ± 16 | 655 ± 28 |
| Elongation [%] | 38 ± 2 | 41 ± 4 |

## 3. Results

In this section, the results from each step of the proposed workflow are presented.

*3.1. Topology Optimization Bounding Box and Constraints*

Starting with the topology optimization step, the resulting mesh that is extracted is shown in Figure 3b.

*3.2. Orientation Optimization*

Then, the mesh is rotated in every possible angle combination, aiming to find the one with the minimum support volume required. The optimal orientation that occurs from the orientation optimization process is shown in Figure 4a. The mesh is rotated by 180° around the *X*-axis and by 82° around the *Y*-axis with respect to the starting orientation.

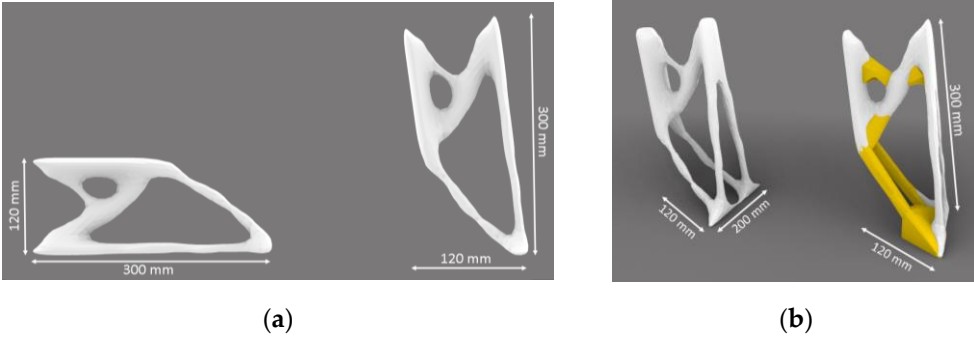

(**a**) (**b**)

**Figure 4.** (**a**) Mesh before (**left**) and after (**right**) orientation optimization; (**b**) mesh before (**left**) and after (**right**) material addition.

### 3.3. Material Addition

The next step is that the remaining overhanging areas are concealed with the suitable geometries generated from the material addition process described in the previous chapter. Figure 4b shows the fully optimized mesh next to the one that occurred from the orientation optimization procedure.

### 3.4. Slicing and Machine Code Generation

As the last step, the fully optimized mesh is sliced (Figure 5a), and the machine code to begin the manufacturing is exported. In this particular case, G-code is used; however, the method would be the same regardless of the specific machine programming language.

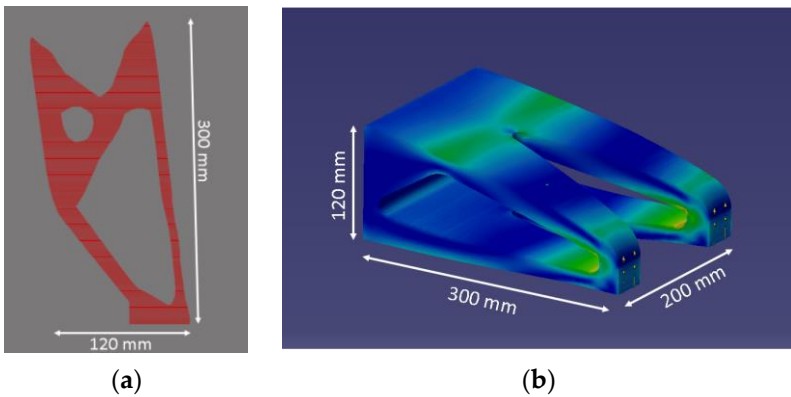

(**a**) (**b**)

**Figure 5.** (**a**) Contoured mesh (front view); (**b**) conventionally designed component.

Following the description of the proposed method to improve the manufacturability of a component through design optimization is the analysis of its effectiveness and evaluation of its results. In order to measure the effectiveness of the recommended workflow, a comparison with a conventional workflow of designing the same component is made. The parameters that will determine the evaluation procedure of the two workflows are the volume of the final meshes, the support volume of the final mesh, and the manufacturability time for a specific AM process. The values of the volumes are measured using Grasshopper 3D, while the manufacturing time is calculated using a commercial CAM software for AM (Ultimaker Cura), based on the specifications of a typical DED three-axis machine (Table 1).

The conventional workflow starts by importing the boundary domain in CAD software (in this case Dassault Systemes CATIA), and after applying the load case and the support conditions, a structural simulation is executed. The results of the simulation are then utilized to manually remove material from the initial domain in an iterative manner until the final component occurs (Figure 5b). The different colors on the part represent the different values of stress that are applied concerning the load and support scenario mentioned above. A blue color corresponds to zero value of stress, and as the stress value

increases, the color changes to green, then yellow, and at last red, which represents the highest stress on the part.

The results of the conventional workflow and each sub-process of the proposed workflow concerning the aforementioned parameters are presented in Table 3.

**Table 3.** Comparison table between the parameters of the two workflows.

| Parameter | Conventional Workflow | Proposed Workflow | | |
|---|---|---|---|---|
| | | Step 1: Topology Optimization | Step 2: Orientation Optimization | Step 3: Material Addition |
| Mesh volume [mm$^3$] | 2667 | 507 | 507 | 723 |
| Support volume [mm$^3$] | 2723 | 654 | 103 | 40 |
| Total volume [mm$^3$] | 5390 | 1161 | 610 | 763 |
| Build time [min] | 13 | 6 | 10 | 9 |

## 4. Discussion

Taking into consideration the presented results from the conventional workflow and those that occur from the material addition sub-process, we can clearly come to the conclusion that the proposed method offers significant advantages regarding its effectiveness compared to the conventional one.

Regarding the results deriving from the conventional workflow and those from the topology optimization process, the difference is obvious. A total volume and build time reduction of around 80% and 50% is observed, respectively. Furthermore, it has to be noted that the conventional workflow includes an iterative process of manually (re)designing and simulating in order to minimize the part's volume, a procedure which is highly dependent on the designer experience, both in terms of the performance of the obtained part as well as the time required to design the part. On the other hand, the topology optimization process is completely automated, the results are consistent, and the whole workflow requires only a few minutes of runtime on an average PC.

In addition, it is evident that utilizing the proposed method significantly reduces the required support volume. This leads to a significant reduction in material usage but also in build time, as supports do not have to be deposited. What is not clearly presented in the above table, but is of equal significance, is the time savings that can be obtained due to the minimization or even elimination of the required subtractive post-processing towards removing support structures, which the proposed method eliminates. This results in a further reduction in the total time required to produce the parts, but also has direct cost reduction implications due to the elimination of cutting tools as well as of an additional process step (no setup time and no additional equipment needed).

The software tools used to implement the proposed method are Rhinoceros 3D and Grasshopper 3D due to their capability to combine a visualization environment and a simple visual programming language which also allows the user to embed their own custom code if needed (using Python or C#), combined with their affordability. The implementation of the proposed method can also be developed using other software tools, while the use of a programming language that includes 3D geometry libraries is proposed.

A limitation of the present work is the usage of bulk material properties in the topology optimization step. To obtain more realistic results, material coupons from the DED process to be used need to be characterized. In addition, actual manufacturing trials of the optimized component need to be completed, and the performance of the AM optimized part needs to be compared to the predicted values. However, this was beyond the scope of this particular work, which focuses on the development of a method that tackles manufacturability limitations by enhancing the design of a component that occurred from the topology optimization process, and not the verification of the topology optimization itself. Nevertheless, the authors intend to carry out such validation tests as a part of future work.

## 5. Conclusions

In this work, a process that optimizes the design of a given component to enhance its manufacturability for a specific AM procedure was presented. This workflow, in order to be accomplished, consists of certain steps which are the topology optimization, the orientation optimization, the material addition, and, to complete this process, a step that slices and generates machine code is also added. The results of this workflow regarding the mesh and the support structures' volume, as long as the estimated build time is compared to that of a conventional workflow, where the design engineer optimizes the component manually, are given as the outcome of a structural analysis. For the specific case study examined in the manuscript, a total volume and build time reduction of around 80% and 50% is observed, respectively, compared with a conventional workflow.

For the proposed method and tool, we have a direct practical implementation in mind, and they could find application in the automotive, aerospace, and architectural industries, where the need for optimized organic designs is constantly increasing, while optimizing the manufacturability is a key factor concerning decision making for production.

Future work will include validation tests to obtain realistic material properties, as well as comparing the predicted and achieved properties of the part through an application of this method in real parts. In addition, the proposed tool is to be improved and developed further so that it can produce a variety of components and geometries and be able to take into account additional AM manufacturability restrictions. Another goal is to upgrade the slicing and machine code generation sub-process to accommodate AM processes that can take advantage of five degrees of freedom (DoF) deposition.

**Author Contributions:** Conceptualization, P.S. and H.B.; methodology, H.B. and M.A.T.; software, M.A.T.; validation, M.A.T.; writing—original draft preparation, M.A.T.; writing—review and editing, H.B. and P.S.; supervision, P.S. All authors have read and agreed to the published version of the manuscript.

**Funding:** This research was co-funded by the European Regional Development Fund of the European Union and Greek national funds through the Operational Program Competitiveness, Entrepreneurship, and Innovation, under the call RESEARCH—CREATE—INNOVATE (project code: T2EDK-03896).

**Data Availability Statement:** Data sharing not applicable.

**Conflicts of Interest:** The authors declare no conflict of interest.

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
