# Peer review of "Manufacturability-Based Design Optimization for Directed Energy Deposition Processes"

_machines, doi:10.3390/machines11090879_

Round 1

Reviewer 1 Report

The paper investigates the optimization of the design of the digital proposal to improve and make the production of parts more efficient by additive manufacturing technology, specifically Directed energy deposition.

The manuscript is well-organized overall, and the topic is related to the interest of this journal. However, before this manuscript becomes acceptable, the authors are required to address the following comments well.

1) The abstract should provide a more concise and focused summary of the study, highlighting the main findings and their significance. For example: explain the abbreviation KPI. Describe the methodologies used in the analyses. Follow the template for writing the abstract in the given journal.

2) In the introduction, specify the goals of the article and the expected benefits of the given issue in more detail.

3) Figure 2 is not mentioned in the manuscript.

4) In Chapter 2.4. and lines 186-190 states that the manufacturing parameters are very important in this step. The production parameters are not mentioned anywhere in the manuscript. It would be appropriate to supplement the parameters of the production process.

5) Please specify exactly what grade of steel was used (line 205). Add the chemical composition of the given steel. State the basic mechanical properties of mentioned steel (YS, UTS, and elongation).

6) In Figure 3(b), insert the scale. Expand the information about the given mesh grid in Chapter 3.1.

7) What is the basis for this statement (lines 201-202): "The optimal orientation of the mesh is rotated by 180° around the X axis and by 82° around the Y axis (Figure 4 (a))."? In Figure 4, it is difficult to observe the optimization measures. It is necessary to revise Figure 4 and highlight the optimized parts. Figure 4 should include a scale.

8) In Figure 5(a), the entire part of the mesh grid is not visible. Remove the white stripe at the bottom of the figure. Figure 5(b) should have the indicated scale bar.

9) The discussion section should go beyond a simple summary of the results and provide a critical analysis and interpretation of the findings, including their implications for future research and practical applications. The discussion as a whole should be elaborated more deeply and individual results should be quantified.

10) Please verify the achieved results with a real manufactured part using DED technology. How is it guaranteed, that the manufactured parts will have the same or higher mechanical properties after a given optimization? This would also confirm the statement in lines 261-265.

11) The conclusion should contain specific research results and not just general findings.

12) The manuscript should be proofread and edited carefully to ensure clarity, accuracy, and consistency in language and formatting.

Based on the abovementioned comments, this manuscript is recommended for major revision. A revised version is required.

Reviewer 2 Report

This study aims to optimize component design by addressing overhanging geometries and support needs, resulting in improved performance, design flexibility, and lighter structures. The proposed workflow significantly reduces part volume, support volume, and build time, demonstrating clear advantages over conventional methods. The paper is well written and  may interest the readers of Machines.
I recommend its publication after minor revisions :

1 -Fig. 3 b) => Add a scale to the figure.
2- Fig. 4 a) b) => 
Add a scale to the figure.
3- Fig. 5 a) b) => Add a scale to the figure.

4- Increase the size of the discussion section a little more.

5- The authors used Rhinoceros 3D and Grasshopper software tools. Can the same proposed methodology be implemented with different software?

Round 2

Reviewer 1 Report

The authors have addressed the comments well. Just correct the following typos:

Lines 217 - 227 should be in the Materials and Methods section and not in the Results section.

If the mechanical properties of the given steel are taken from the literature, please mention the source (line 226, Table 1). It is not clear how the material was processed and thus whether the given mechanical properties are adequate (rolled, HIP, AM technology, heat treated). It is important to use material in the work that is intended for the given technology.

The process parameters in Table 2 should be in the Materials and Methods section and not in the Results section.

In Figure 5(b), explain the color scale (blue, green, and red color).

Based on the abovementioned comments, this manuscript is recommended for minor revision. A revised manuscript is required.
